# Characterization of Nile Tilapia (*Oreochromis niloticus*) Farming Intensities in Liberia

**Mandela Klon-Yan Hinneh** [1,2,*] **, Mbeva D. Liti** [1] **and Geraldine Matolla** [1]

1   Department of Fisheries and Aquatic Sciences, School of Natural Management, University of Eldoret, Eldoret 30100, Kenya; davidmbevaliti@gmail.com (M.D.L.); gmatolla@yahoo.com (G.M.)
2   Department of Fisheries and Aquatic Sciences, University of Liberia, Monrovia 9020, Liberia
*   Correspondence: mandelahinneh@gmail.com

**Abstract:** Fish farming is a rapidly growing food sector in developing nations. Liberia is an example of a least developed country with a large population facing high poverty levels. This has led to the adoption of aquaculture as one of the most important strategies for solving malnutrition and food security problems. However, since the introduction of fish farming, fish yields have been persistently low. To address the shortcomings in fish yields in Liberia, a study was conducted to provide information on fish farming intensities, types, and quality of feeds used by farmers in the culture of *O. niloticus* in Bong, Lofa, Nimba, and Grande Gedeh counties. Using stratified purposive sampling, 120 farmers were interviewed, and their fish feeds were sampled for proximate nutrient analyses. The results demonstrated that fish farming of *O. niloticus* in Liberia is mostly semi-extensive (81.6%), mainly practiced in paddy, barrage, and earthen ponds. On average, farmers produce 165.7 kg ha$^{-1}$ of *O. niloticus* annually, translating to USD 414.25. Farmers use mixed feeding regimes, comprising farmer-made, kitchen waste, and blended commercial feeds. Farmers, on average, spend 43% of their operation cost on feeds, which makes it unsustainable to maintain semi-intensive systems. The main feed ingredients used by Liberian fish farmers are rice bran, wheat bran, corn, palm kernel, and fishmeal. Crude protein levels in feed ingredients are as follows: rice bran (3.7 ± 1.3%), wheat bran (16.4 ± 1.5%), corn (6.3 ± 1.1%), palm kernel cake (14.8 ± 1.4%), and fishmeal (63.8 ± 1.3%). Crude proteins were low in formulated feeds, ranging from 8–15% CP. From this study, poor yields and the slow growth of *O. niloticus* can be attributed to low-protein diets, rendering farming ventures unprofitable and unsustainable for resource-poor farmers in Liberia.

**Keywords:** aquaculture feeds; nile tilapia; malnutrition; semi-intensive; crude protein

## 1. Introduction

The depletion of marine fish stocks and increasing global food insecurity have fueled the rapid growth of aquaculture systems across the world [1–3]. Aquaculture, which encompasses the rearing of fish and other aquatic organisms [4], is currently the fastest-growing food sector in the world [5,6], such that in a span of 15 years from the year 2000 to 2015, production rose from 41,724,569.75 to 106,004,183.75 metric tons, with a whopping 154% growth [7]. Despite the impressive growth record, Africa only contributed 2.5% to global production, while the least developed sub-Saharan Africa (SSA) nations contributed less than 1.0% [8]. The latter has a large percentage of the human population, estimated at 960 million, considered malnourished [9,10]. Thus, embracing fish farming in these countries is critical in alleviating hunger and widening the income base and, therefore, achieving economic empowerment [11,12]. Generally, there has been a slow adoption of aquaculture in some parts of the African continent [13]. The poor aquaculture productivity, particularly in the poverty-stricken region, is underscored by several factors, including a lack of policy framework [14], weak supportive structures and infrastructure [11], inadequate aquaculture management skills, and most importantly, poor-quality feeds [15].

Aquaculture enterprises have high input demands, with feeds making up more than 50% of the total expenses in fish farming [16]. For a farmer to strike a significant profit in a shorter time, high-quality and nutritionally balanced feeds are paramount [17]. However, the high cost of quality feeds impedes its accessibility by the low-resource-based fish farmers [18]. Furthermore, the low quality and high cost of feeds make the sector less sustainable for large-scale productions, particularly in rural areas. According to [19], most rural farmers settle for low-quality fish feeds sourced from kitchen waste and agro-industry residues. Such feeds not only retard fish growth, lengthen the time to reach market size, and reduce the resilience of fish to bio-physical stress but also degrade the quality of pond water [20].

*O. niloticus* is ranked as the primary culture fish species and is preferred by tropical sub-Saharan Africa (SSA) farmers because of its versatility in feeding and fast attainment of market size, particularly with all-male populations [21]. However, it requires feeds of adequate nutritional balance to achieve the target size within a farming season [21]. According to [22], if the pond is sufficiently fertilized, it can sustain the juveniles for up to 80 days of grow-out, after which formulated feeds are needed to promote rapid growth [23]. The low-quality feeds used by farmers have also been shown to contain high antinutritional factors (ANFs), thus reducing the feed conversion rates and inhibiting growth [24].

The problems of slow up-scaling and out-scaling of fish farming are constrained by feeds and inadequate management skills among farmers, which frustrate SSA farmers [25]. The situation is worsened by political and socioeconomic factors, which compound to disadvantage the aquaculture sector, as in the case of Liberia [26]. The country has access to marine fisheries in the Atlantic Ocean. However, several counties are landlocked and rely on fishery products from those with access to the ocean. For decades, Liberia's population has relied on wild fisheries for their fish protein requirements; however, disruption in supply caused by several factors, among others, civil war, Ebola epidemics, and a decline in wild catches, motivated the adoption of fish farming in landlocked counties to bridge the demand gap and offer high-quality fresh fish to consumers. Despite the multisectoral approach to promoting fish farming, no significant contribution from the sector has been recorded [27]. Among the challenges Liberian fish farmers face, fish feed-related constraints emerge as the main hindrance to optimizing farming activities. Although fish feeds challenges are widely acknowledged, information on the types and quality of feeds utilized by Liberian fish farmers is lacking. Thus, this study was conducted to provide information on the status of *O. niloticus* farming intensities and the various types and quality of feeds used by farmers in Liberia.

## 2. Material and Methods

### 2.1. Description of the Study Area

The survey and collection of commonly used fish feed samples were conducted in the four Liberian counties of Bong, Lofa, Nimba, and Grande Gedeh. The counties were selected due to their landlocked status and have historically been inaccessible to an adequate supply of fresh and quality fish from the coastal regions as shown by study map in Figure 1. Additionally, they also have significant aquaculture activities compared to other inland counties. Other landlocked counties were excluded from this study on the following basis: a small number of farmers from which no adequate comparative sample could be selected and a lack of experienced farmers in *O. niloticus* farming of more than five years.

The survey of aquaculture activities in the four counties was conducted using a mixed structured questionnaire with the cardinal objective of establishing the levels of aquaculture intensity, the types of fish feed commonly used by farmers in Liberia, and the levels of education of the households of fish farmers. Samples of fish feeds were also collected to determine the proximate composition of the feeds. A clustered purposeful sampling methodology was applied to identify fish farmers who participated in the survey. In each county, participants from youth (18–35 years), women, and adult men engaged in

the farming of the *O. niloticus* were identified. Using the sample size formula by [28,29],

$\eta = \frac{p(100-p)Z^2}{E^2}$, a total sample size of 120 farmers was determined, where

$\eta$ is the required sample size;
*p* is the percentage occurrence of a state or condition (50);
*E* is the maximum percentage error required (0.05);
*Z* is the value corresponding to the level of confidence (1.96).

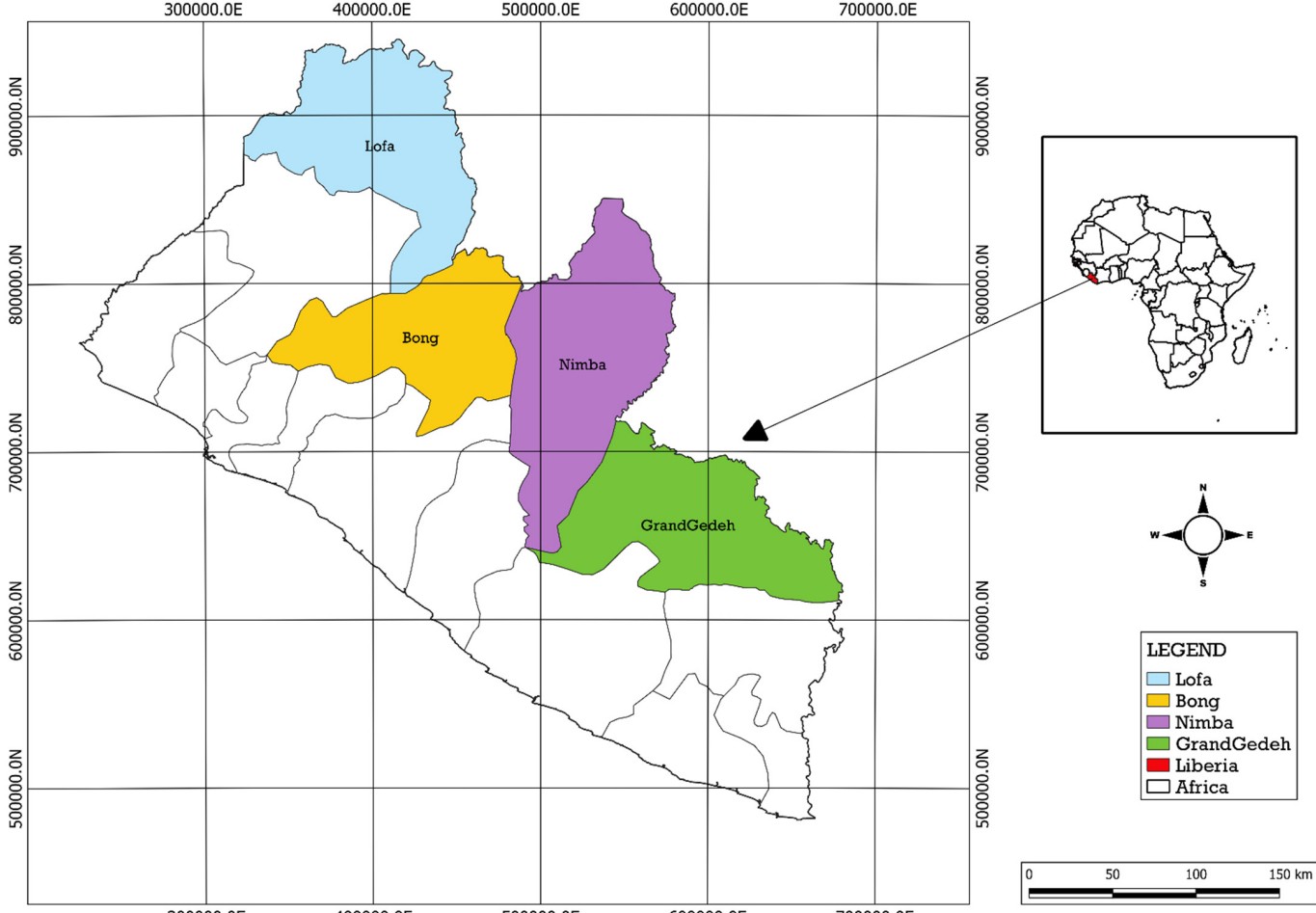

**Figure 1.** Map of Liberia indicating the Bong, Lofa, Nimba, and Grande Gedeh counties of study.

*2.2. Feed Sample Collection and Crude Protein Analysis*

Fish feed samples of 100 g were collected from each fish farmer in the target counties. The collected samples were mainly obtained from locally made feeds, either containing a single ingredient or compounded using more than one ingredient. Feed samples were dried at 60 °C for 24 h and ground to pass through a 1 mm sieve. The total nitrogen content of the samples was determined using the Kjeldahl method [30], and the results were multiplied by 6.25 to obtain the crude protein content, which was expressed as a percentage. Proximate analysis of the feeds was done at the Kenya Marine and Fisheries Research Institute (KEMFRI) and the University of Eldoret laboratories. Data collected were cleaned, arranged, and subjected to descriptive and nonparametric analysis. Significance comparisons were determined at *p* = 0.05 using Kruskal–Wallis (KW) and chi–square tests. All analyses were done using IBM Statistical Package for social science version 23.0, and Microsoft Excel 2016.

### 3. Results

*3.1. Fish Farmers' Characteristics and Demographics*

The study demonstrated that fish farming in Liberia is mainly dominated by farmers of more than 35 years of age, representing 81.6% of the farmers in all the four counties studied. The level of aquaculture is mostly semi-intensive, practiced at 79.2% subsistence level. However, a relatively smaller percentage of farmers, 21.8%, practice semi-commercial fish farming. Pond aquaculture constituted 98%, while cage and tank culture contributed 1.2% and 0.8%, respectively. Gender parity was found to be marked, whereby 75.7% of fish farmers were men; however, skewed gender disparities were found dominant among counties, especially in Nimba and Lofa counties, where less than 20% of women actively participated in fish farming, as shown in Figure 2.

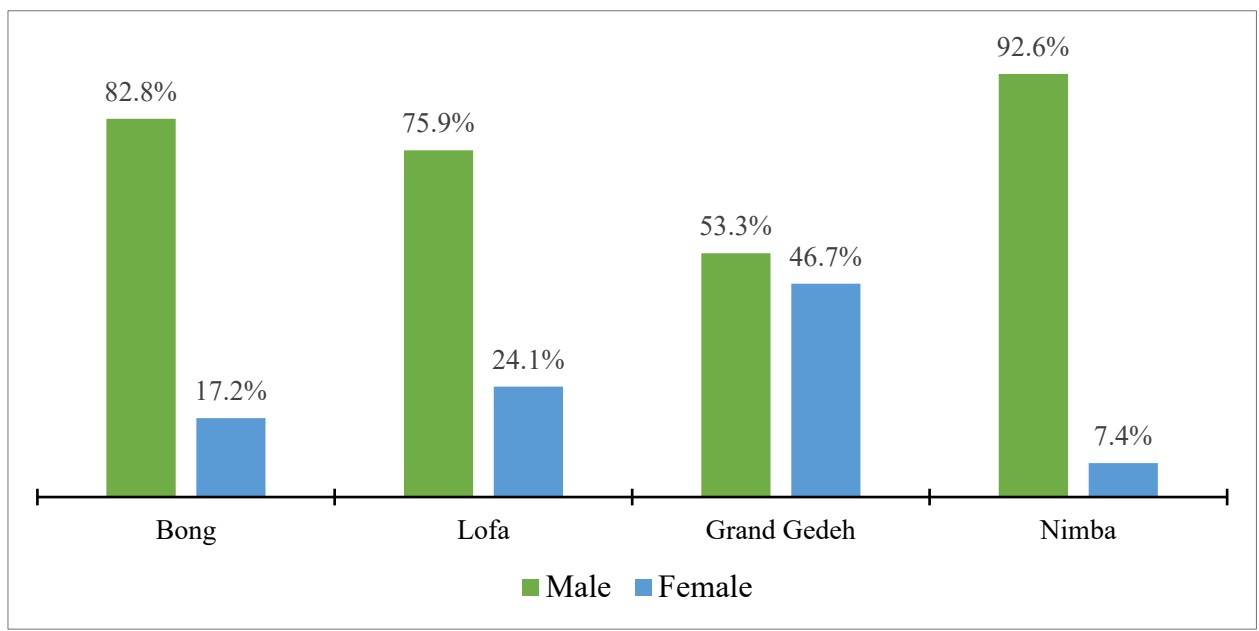

**Figure 2.** Gender composition of fish farmers in Bong, Lofa Grand Gedeh, and Nimba counties, Liberia.

*3.2. Education Level of Fish Farmers*

Less skilled farmers were found to dominate the aquaculture sector, with low basic education and little training in fish farming. The study indicated that 93% of fish farmers had acquired education up to secondary school and below. The literacy level significantly varied among counties, $\chi 2$ (12) = 27.48, $p < 0.05$. Fish farmers residing in Bong and Nimba had acquired a higher level of education and training in aquaculture compared to their counterparts in Lofa and Grand Gedeh, as shown in Table 1.

**Table 1.** Percentage of education level of fish farmers in major farming counties of Liberia.

| County | No Formal Education | Primary | Secondary | Vocational Training | University |
|---|---|---|---|---|---|
| Bong | 31.0 | 37.9 | 27.6 | 3.4 | 0.0 |
| Lofa | 13.8 | 41.4 | 44.8 | 0.0 | 0.0 |
| Grand Gedeh | 16.7 | 46.7 | 36.7 | 0.0 | 0.0 |
| Nimba | 3.7 | 44.4 | 25.9 | 22.2 | 3.7 |

*3.3. Type of Pond Systems Adopted and Fish Species Reared*

Pond type and size are key factors when evaluating the progress and production of aquaculture systems. The study found that barrage and paddy ponds were extensively adopted in Bong, Lofa, and Nimba, with more than 60% of farmers adopting the two as their preferred pond systems, while in Grand Gedeh, 65.5% preferred pit ponds. Few farmers

used concrete ponds, as indicated in Table 2. The pond sizes also varied significantly using the Kruskal–Wallis (KW) test; χ2 (3) = 42.812, *p* < 0.001.

**Table 2.** Adoption (%) of pond types the Bong, Lofa, Nimba, and Grande Gedeh counties of Liberia.

|  | **Paddy Pond** | **Barrage Pond** | **Concrete Pond** | **Pit Pond** |
|---|---|---|---|---|
| Bong | 31.10 | 35.00 | 23.80 | 10.30 |
| Lofa | 33.30 | 25.00 | 14.30 | 20.70 |
| Grand Gedeh | 2.20 | 0.00 | 47.60 | 65.50 |
| Nimba | 33.30 | 40.00 | 14.30 | 3.40 |
| Average size (m$^2$) | 1531.1 | 1503.7 | 478.6 | 710.3 |
| Mean ranks | 71.89 | 81.08 | 25.5 | 44.07 |

Besides rearing *O. niloticus*, farmers also farmed *Tilapia mossambicus*, *Tilapia zilli*, *Heterotis niloticus*, and Catfish (*Clarias gariepinus*), as reported by 21.1%, 26.7%, 45.6%, and 6.7% of farmers, respectively. The species were reared under polyculture systems, rice co-culture, or monoculture in semi-intensive systems. Other species included silverfish (*Oreochromis niloticus* L.), which was integrated into many aquaculture systems in Grand Gedeh County. Table 3 shows different fish types reared by Liberian *O. niloticus* farmers.

**Table 3.** Fish species contribution (%) to fish farming in Bong, Lofa, Nimba, and Grande Gedeh counties.

| Counties | *Clarias gariepinus* | *Tilapia mossambicus* | *Oreochromis niloticus* L. | *Tilapia zilli* | *Heterotis niloticus* |
|---|---|---|---|---|---|
| Bong | 11.5 | 11.5 | 0.0 | 23.1 | 61.5 |
| Lofa | 0.0 | 26.1 | 0.0 | 13.0 | 60.9 |
| Grand Gedeh | 0.0 | 9.1 | 31.8 | 22.7 | 36.4 |
| Nimba | 15.8 | 42.1 | 0.0 | 52.6 | 15.8 |

### 3.4. Influence of Fish Pond Size on Yield of O. niloticus

The relationship between pond size and *O. niloticus* yield is shown in Figure 3. Pond size positively and significantly correlated (R-squared = 0.72, *p* = 0.001) to the production of *O. niloticus*. Big-sized ponds tended to have higher fish yields compared to small ponds.

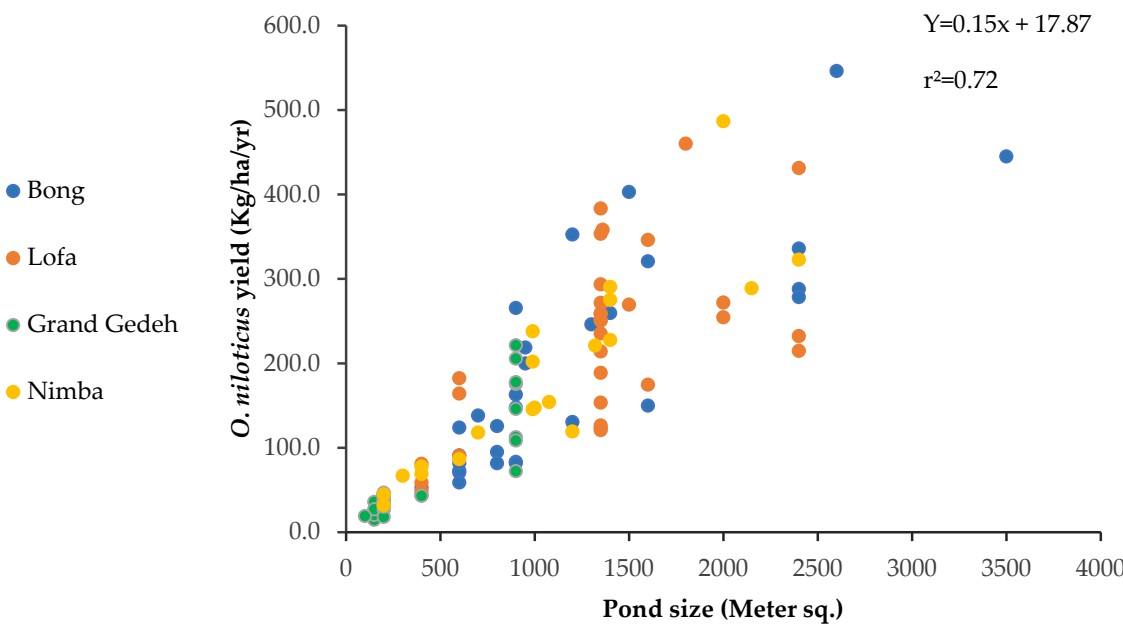

**Figure 3.** Influence of pond size on production of Nile Tilapia Bong, Lofa, Nimba, and Grande Gedeh counties in Liberia.

*3.5. Types of Fish Feeds Used by Farmers in the Production of O. niloticus*

Liberian fish farming can be categorized as semi-intensive, where fish farmers use on-farm-made feeds, including kitchen wastes and locally made feeds. Imported feeds are mainly sourced from Ghana and Sierra Leon. Locally made fish feed was the main type utilized by 67.2% of farmers, while only 32.8% of farmers used blends of imported feeds in their fish feeds. Fish feed types used by farmers per county is indicated in Table 4.

**Table 4.** Types of feeds (%) used in *O. niloticus* farming in Bong, Lofa, Nimba, and Grande Gedeh counties.

| County | Local Made | Type of Feeds Used Household Left Overs | Imported |
|---|---|---|---|
| Bong | 80 | 44 | 60 |
| Lofa | 48 | 56 | 36 |
| Grand Gedeh | 56.7 | 63.3 | 70 |
| Nimba | 61.9 | 52.4 | 57.1 |

The study also demonstrated that farmers used agro-industry-sourced by-products as their main feeds. The feeds were either fed to fish as a single ingredient or blended (formulated). The commonly used feeds included rice bran, palm kernel, wheat bran, corn, fishmeal, and associated blends. Both single and blended feeds had significantly varying crude protein levels in each county investigated. For instance, rice bran, an extensively utilized fish feed ingredient in all the counties under investigation, was found to have an average crude protein content of 3.7%, 4.0%, 2.9%, and 2.3% in Nimba, Bong, Grand Gedeh, and Lofa, respectively. Other feeds collected from the farmers and their respective %CP are indicated in Table 5.

**Table 5.** Levels of crude protein (%) in fish feeds.

| Feed Type | Feed Ingredient | Mean % Crude Protein (Mean ± Standard Deviation) | Range |
|---|---|---|---|
| Single-ingredient feeds | Rice bran | 3.7 ± 1.3 | 2.3–5.7 |
| | Wheat bran | 16.4 ± 1.5 | 14.0–18.1 |
| | Corn | 6.3 ± 1.1 | 5.0–6.8 |
| | Palm kernel cake | 14.8 ± 1.4 | 12.9–16.4 |
| | Fishmeal | 63.85 ± 1.3 | 62.5–65.4 |
| Blended feeds | Ricebran + Soybean | 8.264 ± 5.4 | 3.4–16.4 |
| | Ricebran + Cowpea | 9.45 ± 3.9 | 4.6–14.2 |
| | Ricebran + Corn | 6.99 ± 2.6 | 4.4–11.3 |
| | Ricebran + Fishmeal | 15.27 ± 10.6 | 5.1–32.2 |
| | Ricebran + Palm kernel cake + Fishmeal | 19.34 ± 7.8 | 10.2–28.3 |

*3.6. Fish Feeding Challenges*

The study identified high cost as the major constraint to the use of high-quality commercial fish feeds. Despite farmers adopting extensive and semi-intensive systems, feeds still accounted for 42.98% of total input costs. In addition to the high cost of quality feeds, the spread of disease and pollution of fish environments were ranked by 18.3% and 31.3%, respectively, by farmers.

## 4. Discussion

*4.1. Fish Farmer Characteristics*

The results presented herein demonstrate low education levels among farmers in the fish farming sector in Liberia. These findings are expected as the country has the lowest literacy levels in Africa, attributed to the long period of war that disrupted and decimated the education system for more than 14 years from 1989 to 2003. According to [31], the average literacy in Liberia as of 2020 was 48.3%, and the value dropped down to an average

of 4% for the rural population with upper secondary school level. These results indicate that the fish farming sector is dominated by farmers who have attained primary or secondary education. *O. niloticus* farming is also dominated by mature people of more than 35 years of age, which is explainable by the high resource and time demands required by the venture. Young Liberians' have not accumulated enough resources to invest in businesses, including fish farming. Similar observations were made by [32]. They demonstrated that fish farming is an expensive undertaking requiring surmountable resources and high-level management skills to operate, which many youths in developing countries lack. On the other hand, mature or older farmers adopt fish farming as their old age investment, as reported by [33]. Moreover, older farmers are in a better position to access bank financing compared to the youth.

### 4.2. Pond Characteristics and Influence on O. niloticus Production

The pond fish culture is the main *O. niloticus* mode of farming in Liberia, and this type of aquaculture is common in other sub-Saharan countries [12,34]. The current study established that the adoption of ponds varied by Counties. Nimba, Bong, and Lofa counties use large-sized semi-intensive earthen pond types, including barrage and paddy ponds. In contrast, Grand Gedeh *O. niloticus* farmers have embraced small pond farming mostly in concrete and pit ponds. The variation in types of ponds adopted across the counties could be explained by the fact that fish farming in Liberia originated in the former three counties. Hence, farmers settled for multipurpose, big-sized ponds that were affordable to construct, as in the case of paddy ponds as found by [35]. One interesting finding revealed by the current study is that *O. niloticus* yields increased proportionally to the size of the ponds. A possible explanation for this finding is that the size of the pond is essential in regulating water quality, and in small ponds, water deteriorates faster and has a low recovery rate.

Similarly, small ponds have less oxygen available to the fish, hence compromising growth. Big ponds such as Barrage and Paddy ponds offer stable water quality [36], therefore, quality characteristics such as ammonia, dissolved oxygen, and pH are constantly regulated [37]. When well fertilized, the barrage and paddy ponds will have high plankton levels, the primary fish food [34].

### 4.3. Feed Types, Quality, and Challenges

*O. niloticus* farming in Liberia is mainly semi-intensive, utilizing locally available agro-industry waste products such as rice bran. More than 60% of farmers in each county depend on the by-products of milling as the primary feed for their fish, and this dependency is influenced by cost. For instance, brans of different cereals are cheap and easily accessible to farmers, increasing their usage. Refs. [2,11] reported that more than 80% of fish farmers rely on locally sourced feeds dominated by cereal brans. *O. niloticus* farmers also use imported feeds to complement the locally sourced feeds. More than 40% of farmers in each county blend imported feed with the locally sourced products, surpassing the use of locally sourced feeds in some counties. The findings are supported by findings by [16], who found that fish farmers in Benin highly depended on imported feeds because of the insufficient supply of feeds by the local system. The results indicated that, on average, all agro-industry cereal brans and other plant-based by-products utilized by farmers are significantly lower in crude protein compared to findings by [38], who reported 9.3%, 13.1%, and 15.5% of crude protein in rice bran, maize bran, and palm kernel, respectively. These values are three times higher than those found in the present study. The low quality of feeds in Liberia can be attributed to adulterations, which include the addition of sand and other nutritionally poor materials [11,38]. The fish feed industry in Liberia is poorly developed. Currently, no industry is engaged in manufacturing commercial fish feeds in the country. Imported feeds are very expensive, which hinders their use in Liberia; as [39] explained, imported feeds are highly taxed, hence transferring the cost to the resource-constrained farmers with low purchase capacity.

## 5. Conclusions

The present study demonstrates that *O. niloticus* farming in Liberia is constrained by farmers' inadequacies and the poor quality of feeds. The bulk (79%) of the Liberian fish farmers are in semi-intensive subsistence farming.

## 6. Recommendations

The aquaculture government and nongovernmental organizations in Liberia should strive to improve the quality of fish feeds and introduce best management practices in fish farming through capacity building.

**Author Contributions:** M.K.-Y.H.: development of the research concept, data collection and analysis, and manuscript writing. M.D.L.: research concept, data collection tools, and manuscript proofreading. G.M.: research concept, manuscript writing, and proofreading. All authors have read and agreed to the published version of the manuscript.

**Funding:** This research was funded by Regional Universities Forum for Capacity Building in Agriculture (RUFORUM) grant number [RU/2020/GTA/DRG/005].

**Informed Consent Statement:** Informed consent was obtained from all subjects involved in the study as indicated by the attached survey questionnaire form attached as Appendix A.

**Data Availability Statement:** The data collected under this research are available from the corresponding author upon request.

**Acknowledgments:** The authors are highly indebted to the Regional Universities Forum for Capacity Building in Agriculture (RUFORUM), and Borlaug Higher Education for Agriculture and Research Development (BHEARD) who funded the studies and research. We are also grateful to all research assistants especially James Mumo Mutio of University of Eldoret for his efforts and dedication to data management. We advance our thanks to Liberia fish farmers who willingly participated in the study by interviews and providing samples.

**Conflicts of Interest:** The authors declare no conflict of interest.

## Appendix A. Annex 1: Questionnaire form Used in Data Collection

**Evaluation of Nile Tilapia farming intensities and the impact of different types of feed on Nile Tilapia farming in Liberia**

**Consent statement**

Greetings,

My name is. . . . . . . . . . . {Name of the enumerator}, and I am grateful for your warm welcome. I am here to collect data on your fish farming activities, particularly on feeding, feed types and challenges on production of Nile Tilapia. The data are meant to provide a highlight into research on "Nile Tilapia production intensities in Liberia" by a PhD student who is undertaking his studies at the University of Eldoret, Kenya. Your participation (fully or partially) is voluntarily. The collected data will be treated with utmost confidentiality and will only be used for the stated purpose and the creation of awareness that might help in developing supportive policies towards more efficient fish farming in Liberia. Upon consent to participate, the survey also assumes that (1) You (herein denoted as respondent) are not under the influence of any substance, person/s, or mental-related illness that might interfere with the authenticity of the information you are expected to provide. (2) The responses that you will provide will be consciously made and accurate. Where you find it difficult to answer a question, kindly request for further explanations. I hereby request you to participate in the survey. The interview might take 1 h. **WELCOME**

Consent given        Yes      ☐ No      ☐ (Tick according to respondent's answer)

*{If the respondent declines consent, record the questionnaire number, thank them, and move to another farmer as per the provided list}*

**Section 1: Farmers information**

1     Farmer code...... (to be provided by enumerator).
2     GPS

    (a)     Northing.............................................................................
    (b)     Easting..............................................................................

3     County of the respondent     (Select where applicable)

    i.     Bong     ○
    ii.     Lofa     ○
    iii.     Nimba     ○
    iv.     Grande Gedeh     ○

3.1.     Sub-county of the respondent...........................................
4.     Sex of the respondent (Owner of the fish farm)     (Select where applicable)

    i.     Male     ○
    ii.     Female     ○
    iii.     Prefer not to say     ○

5.     Age of the respondent (Owner of the fish farmer)     (Select where applicable)

    (a)     Below 35 years     ○
    (b)     Above 35 years     ○

6.     Highest level of education attained     (Select where applicable)

    i.     No education     ○
    ii.     Primary     ○
    iii.     Secondary     ○
    iv.     Vocational     ○
    v.     University/College     ○

**Section 2: General fish farming information**

1.     How long have you been practicing fish farming?...... (indicate the answer in years)
2.     What motivated your ambition to start fish farming? (Tick all that applies)

    i.     Source of income     ☐
    ii.     Create employment     ☐
    iii.     Market availability     ☐
    iv.     Diversify investment     ☐
    v.     Availability of government/NGO support     ☐
    vi.     Past experience     ☐
    vii.     Any other     ☐

    State any other factor that motivated you to join fish farming ... ... ...

3.     Total pond size owned............................... (square meters)
4.     (a)     Name fish species you actively farm (Tick all that applies)

    i.     Nile Tilapia     ☐
    ii.     Silver Tilapia     ☐
    iii.     Tilapia Zilli     ☐
    iv.     Tilapia Mossambicus     ☐
    v.     Heterotis niloticus     ☐
    vi.     Catfish     ☐

    State any other species farmed.............................................................

(b)     What are the advantage of farming Nile Tilapia over other species?

    i.     Short maturation period     ☐
    ii.     Efficient feed conversion     ☐
    iii.     Can feed on any type of feed     ☐
    iv.     Can survive under diverse range of environmental conditions     ☐

     v.        Has ready market  ☐
     vi.      Any other  ☐

    State any other Nile Tilapia advantages………

(c)    What are the disadvantages of farming Nile tilapia over other species?

     i.        Feed requirement    ☐
     ii.      Over population  ☐
     iii.     Lack of market    ☐
     iv.     High competition from other species  ☐
     v.        Poor adaptation to Liberian climate  ☐

(d)    What was your total harvest of Nile Tilapia in 2020…… (convert to kgs)
(e)    Rate the productivity of Nile Tilapia over other farmed species

     i.        Very poor  ○
     ii.      Poor         ○
     iii.     Average    ○
     iv.     Good        ○
     v.        Excellent  ○

**Section 3: Fish feeding**

1.0    What type of feed do you use to feed your fish?

     i.        Local commercial feeds  ☐
     ii.      Imported commercial feeds  ☐
     iii.     Kitchen remains       ☐

2.0    Are feeds used in {Section 3, (1)} continuously available for production period?
    Yes  ☐        No  ☐

3.0  (a)    If you use local formulated fish feeds, do you prepare the feeds yourself?
    Yes  ☐        No  ☐
    (b)    If Yes, which ingredient do you use and their combination ratio?…………
    (c)    Where do you source your feed ingredients?

     i.        On-farm ingredients (crops, grains, etc.)  ☐
     ii.      From food processors (millers, etc.)      ☐
     iii.     Any other?                ☐

    Specify other sources for your feed ingredient

    (d)    If you don't produce your own feeds, where do you source farmer-formulated feeds?

     i.        Other fish farmers  ☐
     ii.      Local feed vendor  ☐
     iii.     Local market      ☐
     iv.     Any other        ☐

    Specify any other…………………………

    (e)    How much do you pay per kilograms of locally formulated feeds…… (state amount in USD)
    (f)    If you use commercial feeds, where do you source them? (Tick all that apply)

     i.        Local agro-dealers  ☐
     ii.      Local markets      ☐
     iii.     Government       ☐
     iv.     Nongovernmental organizations'  ☐
     v.        Import     ☐
     vi.      Other      ☐

    (g)    What challenges do you face in accessing commercial feeds?

     i.        Expensive     ☐

ii.      Not available locally   ☐
iii.     Poor quality   ☐
iv.     Others   ☐

Specify other challenges………………………………

4.0   How do you administer the feed to the fish?

i.      Manual broadcasting   ○
ii.     Automated feeding   ○
iii.    Other          ○

Specify any other feeding mechanism used………………………

5.0   (a)    Do you keep records on the impact of feed on the growth of the fish? Yes ☐     No ☐

       (b)    If yes, what are the growth aspects monitored? (Tick all that apply)

i.      Fish weight    ☐
ii.     Fish length    ☐
iii.    Fish yield     ☐
iv.    Survival rate   ☐

6.0   (a)    Have you received training on how to formulate feed for the farmed fish species? Yes ☐ No ☐

       (b)    If **YES**, what aspect of feed formulation were you trained in? (Tick all that apply)

i.      Feed rationing     ☐
ii.     Feed ingredients    ☐
iii.    Feed types        ☐
iv.    Feeding different fish species   ☐

       (c)    Who provided the training?     (Tick al that apply)

i.      Government extension officers     ○
ii.     Nongovernmental organizations    ○
iii.    Research institutions (universities, agriculture organizations, etc.)   ○
iv.    Fellow farmers   ○
v.     Any other      ○

Specify any other place/organization where you received training…..

7.0   What are some of challenges you face in feeding your fish?
8.0   What management practice do you adopt to ensure that the fish are in the best of condition?
9.0   What are your recommendations on improving aquaculture in the country?

i.      Increased government support in terms of inputs   ☐
ii.     Improved extension services     ☐
iii.    Improved access to quality feed   ☐
iv.    Other                   ☐

Specify any other suggestion…….
Finally, you are requested to provide samples of your fish feeds to the enumerator for laboratory testing. (Provide at least 500 g of each fish feed type).
**Thank you for your participation**

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
