# Peer review of "Characterization of Nile Tilapia (Oreochromis niloticus) Farming Intensities in Liberia"

_2673-9496, doi:10.3390/aquacj2030011_

Round 1
Reviewer 1 Report
The manuscript presents an interesting characterization of fish production in Liberia.
However, some aspects of the writing and presentation of results need to be improved, as follows:
1) A more qualified review of the English language is required.
2) Suggestion to change the title: "Characterization of Nile tilapia production in Liberia". The characterization of the diets and the intensities used were variables studied and do not need to be mentioned in the title.
3) I suggest including the form used to interview producers as an annex to the study.
4) The results of some statistical analyzes were presented. However, the M&M section did not describe which analyzes were used and which criteria were used to only present these analyzes for some variables.
5) Why only analyze the crude protein of the diets?
6) Line 157: put the scientific name right after the citation of silverfish;
7) Table 3: standardize the presentation of species by common or scientific name. If using the common name, insert the species as a footnote.
8) Figure 3 - the r2 value is partially visible. To correct.
9) Figure 4 - I suggest the authors review how to present these data. In the form of percentage of producers, the information is confusing. Apparently, more than one of the feed classes presented were used by the same producer in the counties, since the sum of the classes exceeds 100%. As it stands, this information on the graph is confusing.
Author Response
Thank you for the positive criticism and suggestions that refines this manuscript—my responses to your concerns.
1) A review of the English language has been done
2) Title changed to "Characterization of Nile tilapia (Oreochromis niloticus) Farming Intensities in Liberia"
3) Questionnaire form used for data collection attached
4) Added the method of data analysis (descriptive and non-parametric tests like Kruskal Wallis and Chi square comparison tests.
5) Protein is the most essential part of fish Nutrition. It’s the first crucial nutrient in Fish Nutrition. Also, it’s the most expensive component in fish Nutrition. Protein is the first limiting and most expensive nutrient in fish production
6) Line 157 corrected as per the suggestion
7) Table 3. names converted to scientific names
8) The visibility of r2 improved
9) The figure represents multiple responses. In this case, a farmer can be using different types of feeds not only limited to one. Thus, the percentage may not add up to 100%
Reviewer 2 Report
The paper is important as it deals with a subject of interest for the production of food for a needy population. This study is a map of tilapia production in the country and tries to point out the deficiencies found. However, to improve the publication, some points must be clarified.
Abstract
All scientific names must be written in italics.
Line 16. Review: .... O. niloticus. In Bong.......
Check the spacing between words. In fact, this must be done throughout the text.
Material and methods
In Figure 1 in the map of Liberia, 6 counties are indicated and only 4 were sampled. Placing only the four on the map to avoid confusion.
How many farmers were sampled in each county?
How were the data from the food samples statistically analyzed?
Results
Line 162 3.4. Influence of Fish pond size on yield of O. niloticus
The relationship shown in Figure 3 does not represent the reality. This data must be relative to the stocking density in each pond type. The figure should include kg m-3 and not just kg. Especially when it comes to tilapia that are territorial and the density will interfere in the final production. See lines 221 and 222 (Discussion). Of course, larger ponds can store more fish than smaller ones, but density must be respected. For this reason, the calculation in Figure 3 must be revised to bring it closer to reality. Another item that must also be taken into account is the flooded area of ​​each visited enterprise.
Discussion
Review lines 221 and 222.
Review the citations from line 235.
References
The reference Kaleen and Sabi, 2221 and Okai, 2018 are not included in the text.
The reference Sodyinu et al., 2016 was not found in the reference list.
Author Response
I appreciate the suggestions made and I have effected them, my responses are as follows.
1) Editing the manuscript by the journal system might have removed spaces between some words particularly in Abstract, however, I have corrected
2) All scientific names italicized
3) New map of only four counties attached
4) Crude protein data were subjected to descriptive statistics, statistical comparisons were not be possible because every farmer used a different combination of feed types.
5) There was a mistake in labeling Y-axis and it was corrected to Yield (kg/ha/year)
6) Lines 221 and 222 were reviewed as suggested
7) Added citation to line 235
8) Bibliography and intext citations have been harmonized